# Characterization and Correlation of Dominant Microbiota and Flavor Development in Different Post-Mortem Processes of Beef

**DOI:** 10.3390/foods12173266

**Published:** 2023-08-30

**Authors:** Hengpeng Wang, Jipan Wang, Yinlan Wang, Sumin Gao, Shuangyi Xu, Xiaobo Zou, Xiangren Meng

**Affiliations:** 1Key Laboratory of Chinese Cuisine Intangible Cultural Heritage Technology Inheritance, Ministry of Culture and Tourism, College of Tourism and Culinary Science, Yangzhou University, Yangzhou 225127, China; yzuwhp@163.com (H.W.); yzuwjp@163.com (J.W.); gsumin@163.com (S.G.); xsy11yzu@163.com (S.X.); 2International Joint Research Laboratory of Intelligent Agriculture and Agriproducts Processing, School of Food and Biological Engineering, Jiangsu University, Zhenjiang 212013, China; zou_xiaobo@ujs.edu.cn; 3Engineering Technology Research Center of Yangzhou Prepared Cuisine, Yangzhou 225127, China; 4School of Food Science, Jiangsu College of Tourism, Yangzhou 225000, China; wyl1730@163.com; 5Chinese Cuisine Promotion and Research Base, Yangzhou University, Yangzhou 225127, China

**Keywords:** beef rump, postmortem aging, key volatile compounds, dominant microbiota, correlation analysis

## Abstract

Post-mortem aging could enhance the unique flavors of beef via several biochemical pathways. The microbiota is one of the important factors in the flavor development of aging beef, but their potential relationship has rarely been studied. This study characterized the apparent meat quality, flavor profiles, and microbial communities of beef during the different post-mortem processes, followed by the investigation of the correlations between the dominant microbiota and key volatile compounds. The results showed that wet-aged beef has a higher product yield and more stable color than dry-aged beef, as evidenced by the significantly lower value of aging loss and discoloration (ΔE). According to the odor activity value, 11 out of 65 compounds were categorized as aroma-active components, and 9 of them, including 1-pentanol, 1-octen-3-ol, hexanal, nonanal, heptanal, octanal, 2-nonenal, (E)-, 2-octenal, (E)- and 2-decenal, (E)-, were enriched in beef wet-aged for 7 d. Significant variances were found in the microbial communities of different aging beef. Of these, 20 microbiota (with 10 bacterial and 10 fungal genera) were recognized as the dominant genus. Partial least squares regression combined with a correlation network model revealed that five microbial genera, including *Trichosporon*, *Prauserella*, *Rhodotorula*, *Malassezia*, and *Corynebacterium*, constituted the functional microbiota responsible for flavor formation in aging beef and were positively associated with ≥7 key volatile compounds (*p* < 0.05, |ρ| > 0.7). This study suggests that the application of wet aging within 7 d on beef is better for meat quality and provides novel insights into the mechanisms of flavor formation in post-mortem aging beef via functional microbiota.

## 1. Introduction

Wet-aged beef is most commonly achieved by vacuum packaging carcasses or steaks in a controlled environment to improve palatability and extend shelf life [1]. Dry-aged beef, on the other hand, is unpackaged under more stringent conditions, including temperature, humidity, and air velocity [2]. Although high-end restaurants often market dry-aged beef as a premium product with better flavor, it remained controversial whether the quality of dry-aged beef was definitely higher than that of wet-aged beef [3].

Flavor is one of the most important indicators of the overall acceptability of beef [4]. Compared to freshly slaughtered beef, aged beef usually has a better flavor [5,6]. In general, dry-aged beef has a more beefy, toasty, and nutty flavor, while wet-aged beef may have an intense sour, metallic, and bloody flavor [7]. At present, the flavor formation mechanism of aging beef has received extensive attention. Most studies believe that the oxidation and degradation of fat (the interaction between free amino acids) and reducing sugars are the two main formation pathways of flavors in aging beef [8,9]. In particular, the growth of microorganisms in beef during the post-mortem aging process is inevitable. Some phyla, such as *Proteobacteria*, may contribute to the increased availability of flavor precursors in aged beef [10]. A few bacteria and fungi, including *Thamnidium* sp., *Pilaira anomala*, and *Debaryomyces hansenii*, have been confirmed to participate in the flavor formation of dry-aged beef by releasing exogenous proteolytic and lipolytic enzymes [8,11]. The diversity of microorganisms in aged beef is very complicated. We have speculated that some microorganisms may positively influence the development of the characteristic flavor of aged beef, but their potential correlations are still unclear.

Novel high-throughput sequencing technology can more accurately characterize specific microbial communities within complex ecosystems [12]. The effect of microorganisms on the flavor of meat has been identified in various fermented products, such as ham [13], dry-cured grass carp [14], and sour soup [15]. However, to date, there has been limited research on the microbial components that influence flavor development in post-mortem beef using different aging methods.

In this study, we first characterized the volatile compounds in different aging beef by combining the headspace solid-phase microextraction (HS-SPME) and gas chromatography-mass spectrometry (GC-MS) techniques. In addition, we amplified the V3–V4 regions of bacterial 16S rRNA genes and fungal internal transcribed spacer (ITS) regions to identify microbial communities and, consequently, intensively evaluate the complex microbiota in wet- and dry-aged beef. Finally, partial least squares regression (PLSR) and the correlation network model were used to evaluate the relationship between microbial communities and key flavor compounds. These results will reveal the core microbiota involved in beef with different aging methods and their contribution to flavor formation during the post-mortem aging process.

## 2. Materials and Methods

### 2.1. Preparation of Wet- and Dry-Aged Beef

A total of 18 Northeast Simmental bulls (approximately 24 months old) were slaughtered following the same standard commercial procedures by Wuxi Tianpeng Group Co., Ltd. (Wuxi, China). After slaughter, a total of 36 beef rumps were collected and immediately cooled to ensure that the core temperature dropped below 4 °C. The aging time was recorded as 0 h, then covered with crushed ice and transferred to the laboratory within 1 h. Six randomly selected beef rumps were aged for 1 d, 3 d, and 7 d, respectively, using two aging methods. Wet aging was conducted with a vacuum packaging bag (1.6 mL O_2_/m^2^/24 h, 0.966 g H_2_O/m^2^/24 h); Dry aging samples were packaged in moisture-permeable packaging (280 × 350 × 0.04 mm^3^; 2265 mL O_2_/m^2^/24 h, 7900 g H_2_O/m^2^/24 h). All aging samples were placed in a freezer (4 °C, Haier SC-650HS, Qingdao Haier Refrigeration Equipment Co., Ltd., Qingdao, China) under controlled conditions (airspeed, 2–7 m/s; temperature, 4 °C; humidity, 85 ± 10%). To minimize the effect of different positions on the samples, we changed the position of the samples every 6 h. After each aging period, the beef samples were cut into rectangular blocks weighing approximately 45 ± 1 g (length × width × height: 40 × 30 × 20 mm) and separately vacuum-packed and frozen at −70 °C until analysis.

### 2.2. Apparent Meat Quality Testing

#### 2.2.1. Measurement of pH and Color

The pH of all samples was measured using a portable pH meter (Matthaus pH-STAR, Shanghai ZhenMing Scientific Instrument Co., Ltd., Shanghai, China), and each sample was measured in at least three different locations. In addition, a colorimeter (CR-400, Minolta Co., Tokyo, Japan) was used to record the lightness (L*), redness (a*), and yellowness (b*) values of each meat sample. The ΔE value was calculated using the equation below, which indicates the total color difference between the tested sample and the original sample aged for 0 h.
(1)$$\Delta E=\sqrt{{\left(\Delta L^{*}\right)}^{2}+{\left(\Delta a^{*}\right)}^{2}+{\left(\Delta b^{*}\right)}^{2}}$$

#### 2.2.2. Measurement of Aging Loss

Aging loss was determined using the change in weight of the beef rumps before and after aging. The percentage of aging loss was calculated as follows:(2)$$\mathrm{Aging}\;\mathrm{loss}\;(\%)=\frac{W_{1}-\;W_{2}}{W_{1}}\times \;100\%$$

W_1_: weight before aging (g); W_2_: weight after aging (g).

### 2.3. Volatile Compound Analysis

#### 2.3.1. Gas Chromatography-Mass Spectrometry

The volatile compounds of each sample were analyzed according to the method of AlDalali with minor modifications [16]. The beef sample (2 g) was placed in a 10 mL headspace vial, and 1 µL of internal standard (2-methyl-3-heptanone, 0.816 mg/L) and 1 µL of saturated table salt water were added separately. The vial was sealed with a silicone septum, and 75 μm of polydimethylsiloxane/divinylbenzene coated fiber (Supelco, Bellefonte, Bellefonte, PA, USA) was inserted into the vial for flavor extraction at 80 °C for 50 min. Subsequently, the fiber was introduced into the GC/MS inlet for desorption at 250 °C for 5 min. Chromatographic separation was performed individually on DB-5MS columns (30 m × 0.25 mm × 0.25 μm; Agilent Technology, Santa Clara, CA, USA). The carrier gas was nitrogen, with a flow rate of 1 mL/min. The GC-MS was conducted at an initial temperature of 40 °C for 2 min, ramped up to 160 °C at 5 °C/min, then increased by 10 °C/min to 250 °C and was held for 4 min. The ion source temperature was 230 °C. The mass scan range was 40–450 *m*/*z* at an electron energy of 70 eV.

Qualitative and quantitative analysis: Volatile compounds were confirmed using NIST 14.0 (National Institute of Standard and Technology, Gaithersburg, MD, USA) and WILEY 6.0 database (John Wiley, Hoboken, NJ, USA) in combination with Kováts retention index (RI) calculated from C7–C40 n-alkanes. The concentrations of volatile compounds were determined via semi-quantitative method using internal standards.

#### 2.3.2. Calculation of Odor Activity Value (OAV)

Thresholds for compounds in water were determined based on GC-MS qualitative and quantitative results and relevant literature.
(3)$$\mathrm{OAVx}=\frac{\mathrm{Cx}}{\mathrm{OTx}}$$

OAVx: odor activity value of volatile compound X.

Cx: concentration of volatile compound X (on average) (ug/kg).

OTx: threshold of volatile compound X in water (ug/kg).

OAVx > 1 indicates that this component plays a direct role in the overall flavor.

### 2.4. Analysis of Microbial Community Succession via High-Throughput Sequencing Technology

#### 2.4.1. Sample Metagenomic DNA Extraction

The samples were the lean portion collected from the crust/surface of aged beef rumps (dehydrated surfaces were trimmed from dry-aged beef rumps). First, 3 g sample and liquid nitrogen were placed in a container for complete grinding. Then, 0.3 g of the sample was transferred to a test tube and extracted by grinding in a tissue disruptor. The OMEGA Soil DNA Kit (Omega Bio-Tek, Norcross, GA, USA) was used for DNA extraction. DNA integrity and concentration were quantified using 8% agarose gel electrophoresis and UV spectrophotometer, respectively.

#### 2.4.2. PCR Amplification

The 16S rRNA gene in the V3-V4 region was amplified using primer 338F (ACTCCTACGGAGGCAGCA) and 806R (GGACTACHVGGTATCTAAT), while the ITS1b region was amplified with PCR primers ITS1F (CTTGGTCATTTAGAGGAAGTAA) and ITS2 (GCTGCGTTCTTCATCGATGC). Amplification was performed as follows: PCR was pre-denatured at 98 °C for 30 s to completely denature the template DNA, then the amplification cycle was started. In each cycle, the template was held at 98 °C for 15 s to denature and then cooled to 50 °C for 30 s to completely anneal the primer from the template. The template was held at 72 °C for 30 s to allow the primer to extend over the template and bind tightly to the DNA to complete a cycle. This cycle is repeated 25–27 times to ensure high accumulation of amplified DNA fragments. Finally, the amplification products were held at 72 °C for 5 min to ensure amplification product extension and stored at 4 °C. Amplification results were analyzed via 2% agarose gel electrophoresis, fragments of interest were excised, and the target was recovered using the Axygen DNA Gel Recovery Kit (AXYGEN Biosciences, Union City, CA, USA).

#### 2.4.3. Illumina Sequencing

Qualified libraries were used to utilize the NovaSeq 6000 SP Reagent Kit (500) on Illumina NovaSeq machine cycles) for two-ended sequencing of 2 × 250 bp. The QIIME2 dada2 analysis stream or the vsearch software analysis process were followed for sequence denoising or OTU clustering. The specific composition of each sample (group) at different taxonomic levels of species was developed line display for an overview of the whole picture. According to the distribution of ASV/OUT in different samples, the α-diversity level of each sample was assessed and passed by thinning. The thinning curve reflected whether the sequencing depth was appropriate.

### 2.5. Statistical Analysis

All results were performed in three replicates. Differences between samples were determined via one-way analysis of variance (ANOVA) using SPSS 19.0 (IBM, USA). Origin 2022 software was used to plot the data. Sequence data analyses were mainly performed using QIIME2 and R packages (v3.2.0). Shannon diversity index and Simpson index were calculated using the ASV table in QIIME2 and visualized as box plots. The significance of differences in microbiota structure between groups was assessed via PERMANOVA (permutational multivariate analysis of variance) and ANOSIM (analysis of similarities). Microbial diversity and community composition were analyzed in the Personalbio Gene Cloud (https://www.genescloud.cn/ (accessed on 21 December 2022)). Partial least squares regression was performed using XLSTAT Pro to determine the association between microbiota and flavor, and the correlation network model between the key volatile compounds and dominant microbial communities was visualized using Omicstudio Cloud (https://www.omicstudio.cn/ (accessed on 3 April 2023)).

## 3. Results and Discussion

### 3.1. Changes in Apparent Meat Quality

As shown in Table 1, with increasing aging time, the pH first decreased and then increased, reaching the lowest value (5.70 and 5.62) at 1 d of aging, which was primarily due to the accumulation of hydrogen ions via lactic acid formation, but the increase in pH is accelerated via the deamidation of meat proteins and the accumulation of bacterial metabolites [17]. Compared with the aging time, the aging methods had little effect on the pH of beef (*p* > 0.05). Obviously, the aging loss of wet-aged beef was significantly lower (*p* < 0.05) than that of dry-aged beef regardless of the aging time, which is consistent with the result described by Kim et al. [18], who reported that dry-aged beef had more water evaporation than wet-aged beef.

The lightness value (L*) of WA7 was higher than that of DA7, which may be due to the increase in reflectance spectra for the higher moisture content in wet-aged meat [19]. The redness value (a*) of dry-aged beef within 3 d was higher than that of wet-aged beef but had a significant decrease (*p* < 0.05) after 7 d of aging compared to wet-aged beef, which was mainly related to the large accumulation of metmyoglobin on the surfaces of dry-aged beef with prolonged aging. This result also showed that dry-aged beef could obtain a better red meat color at the early stage of aging compared with wet-aged beef. The change tendency of the yellowness value (b*) was consistent with the redness and reached the highest value at 1 d of dry-aged beef, which could be explained by the higher degree of oxygen penetration into the interior of dry-aged meat [20]. The ΔE value in dry-aged beef was significantly higher (*p* < 0.05) than that in wet-aged beef at the same aging time, which could be due to the severe moisture loss and higher degree of protein oxidation and degradation in dry-aged meat [21,22]. Overall, wet-aged beef has a higher product yield and more stable color compared to dry-aged beef.

### 3.2. Changes in Volatile Flavor Compounds

The volatile compounds of wet- and dry-aged beef were quantitatively determined, and 65 volatile compounds were classified into seven categories according to their structural similarity (Figure 1A). Of these, 10 unique compounds were detected in wet-aged beef, and 20 unique compounds were detected in dry-aged beef. The post-mortem aging process could significantly increase the concentration and amount of flavor compounds (Figure 1B,D), which could be related to the gradual degradation and oxidation of meat protein [12]. A heat map combined with agglomerative hierarchical clustering was used to investigate the quantification of volatile compounds and their relationship to the characteristics of aged beef. As presented in Figure 1C, the higher concentrations of volatile compounds were indicated by darker red squares. It could be identified that the beef wet aging for 7 d had the highest flavor concentration, distinctly different from other aging groups.

As shown in the Sangji map (Figure 2), aldehydes and alcohols were confirmed as the main flavor substances in beef. Most aldehydes are products of lipid oxidation triggered by endogenous enzymes or microbial activity, with low odor thresholds and distinctive odor characteristics [14]. The aldehyde content of wet-aged beef was significantly higher (*p* < 0.05) than that of dry-aged beef. The changes in aldehyde content were mainly influenced by hexanal (grass, tallow, fat), nonanal (fat, citrus, green), heptanal (fat, citrus, rancid), octanal (fat, soap, lemon, green), 2-nonenal, (E)- (green, soapy, cucumber), and 2-octenal, (E)- (roast, fat), which were predominant in WA7 and DA7, especially in WA7 samples. The increase in aldehyde content may be due to the oxidation of polyunsaturated fatty acids and processing conditions [23], which could explain the higher aldehyde content in beef after wet aging for 7 d.

Alcohols are precursors of aging esters and are produced via glucose metabolism and amino acid dehydrogenation [24]. The total alcohol content increased from 23.09 μg/kg to 42.23 μg/kg as wet aging progressed. The dominant alcohols were 1-pentanol and 1-octen-3-ol, both of which reached maximum levels in WA7 samples. 1-Octen-3-ol is an unsaturated alcohol with a strong mushroom aroma that is identified as being produced via the enzymatic or non-enzymatic degradation of linoleic acid [21]. In particular, sweet and fruity esters are formed via the esterification of alcohol with esterases produced via microorganisms [25,26].

Ketones are lipid oxidation products with a low odor threshold and are associated with the fatty flavor of cooked meat [27]. Some ketones are important intermediates in the formation of heterocyclic compounds [28], which may contribute little to the flavor profile of beef. Furans, odor-active volatiles, are derived from the oxidation of fatty acids [29]. The concentration of 2-pentylfuran increased with the extension of aging time in both wet- and dry-aged beef. The proportion of the total ester content of beef was relatively lower than the other types of compounds. Hydrocarbons were detected only in dry-aged beef, which was consistent with the result of Ha et al. [2]. In general, the formation of hydrocarbons is due to the oxidation of lipids. Obviously, dry-aged beef exposed to air has a higher degree of lipid oxidation compared to wet-aged beef. Overall, the results proved that post-mortem aging can significantly improve the flavor of beef. Furthermore, the wet- and dry-aged beef had their own key volatile compounds at each stage of the post-mortem process, and wet-aged beef has a higher degree of flavor enrichment compared to dry-aged beef within 7 d of aging.

### 3.3. Characterization of the Key Volatile Compounds

In this present study, the odor activity value (OAV) was used as an important indicator to identify the key volatile compounds during beef aging. As shown in Table 2, eleven volatile compounds had OAVs greater than the threshold value. 1-Pentanol had the highest average OAV, followed by hexanal, nonanal, and 2-nonenal, (E)-. Moreover, octanal and 1-octen-3-ol were found to contribute significantly to the overall aroma of beef, exhibiting characteristic aroma notes such as mushroom, fatty, grassy, citrus, soapy, honey, apple peel, and fruity [30]. These results confirm the findings of Setyabrata et al. [10]. The key volatile compounds began to form in the early stages of post-mortem aging, and most of the compounds were enriched with increasing aging time, especially in WA7 samples.

### 3.4. Dynamic Analysis of Microbial Communities

#### 3.4.1. Illumina Sequencing

The alpha diversity indices for bacteria and fungi of wet- and dry-aged beef with different aging times are shown in Table 3, including reads, observed Species, Shannon, Simpson, Chao1, and Goods coverage analyses. The results showed that bacteria obtained a total of 643,432 high-quality sequence quantities and 2228 observed species, and fungi obtained a total of 1,018,522 high-quality sequence quantities and 182 observed species. The coverage of all samples is ≥0.99, indicating that the sequencing depth was reasonable and basically covered all species in the samples [31]. In addition, the microbial diversity index of each group of samples was quite different, and the Shannon index of bacteria was greater than that of fungi in all samples, indicating that the beef rump has richer bacterial diversity. Moreover, as shown in Figure 3, based on the results of reaching a plateau of bacterial and fungal Rarefaction and Shannon curves, we find that the species in beef rump did not significantly increase with the number of sequences, and the sample sequencing depth could cover the entire microbial diversity of the samples, further suggesting that the sample sequence is sufficient for data analysis [32].

#### 3.4.2. Composition of Microbial Communities

The microbial community composition of bacteria and fungi of different beef is shown in Figure 4. Based on the results of species annotation, the 10 most abundant species were selected as the dominant microorganisms at the phylum and genus level. As presented in Figure 4A, *Actinobacteria*, *Proteobacteria*, and *Firmicutes* were the three dominant bacteria at the phylum level, accounting for 37.27%, 32.31%, and 22.17% of the relative abundance, respectively. Among them, *Proteobacteria* increased from 21.32% to 63.01% in dry-aged beef for 7 d of aging. The results were in agreement with the report of Capouya et al. [33], who concluded that *Proteobacteria* was the predominant microbial phylum in dry-aged meat. Ongmu et al. [34] thought that *Proteobacteria* is closely related to amino acid and lipid metabolism, which means that the biological metabolism in dry-aged beef is more vigorous. Compared with dry-aged beef, the relative abundance of *Firmicutes* was higher and more stable during wet aging. Ribeiro et al. [35] also reported that *Firmicutes* was more abundant in wet-aged beef.

At the genus level, Figure 4B showed that with the extension of dry aging time, *Corynebacterium*, *Macrococcus*, *Dermacoccus*, *Brochothrix*, and *Kocuria* decreased by 16.63%, 4.61%, 3.69%, 14.50%, and 3.62%, respectively, while *Acinetobacter*, *Rhodococcus*, and *Prauserella* showed an increasing trend. *Corynebacterium* also decreased from 22.14% to 17.67%, while *Macrococcus*, *Enhydrobacter*, *Acinetobacter*, and *Prauserella* showed varying degrees of increase during wet aging. Ahmed et al. [36] reported that *Acinetobacter* could contribute to meat flavor enhancement by hydrolyzing fats and proteins. *Acinetobacter* and *Kocuria* have also been detected in beef jerky [37]. The findings suggest that members of the genus *Acetobacter* contribute to improving the flavor of aged beef.

Regarding the fungal communities at the phylum level (Figure 4C), significant differences between the two phyla were observed in the fungal communities between wet- and dry-aged beef. *Basidiomycota* (53.68%) was the most abundant in CG samples, followed by *Ascomycota* (44.83%). *Basidiomycota* dominated wet-aged beef with an average relative content of 89.87%. During dry aging, *Ascomycota* significantly dominated DA1 (99.92%) and DA7 (99.97%), while *Basidiomycota* was absolutely dominant in DA3 (99.49%). These results were similar to those of Tian et al. [38]. The aging process had a great influence on the species and abundance of fungi. *Candida*, *Cutaneotrichosporon, Apiotrichum*, and *Filobasidium* were well represented at the genus level (Figure 4D). *Candida* was absolutely dominant in DA1 (99.83%) and DA7 (99.92%). *Apiotrichum* dominated WA1 (99.59%) and was also found in CG (28.51%) and WA7 (16.27%). *Filobasidium* was the capital in WA3 (78.71%). *Cutaneotrichosporon* was most abundant in WA7 (60.22%) and DA3 (99.48%).

#### 3.4.3. Similarity of Microbial Communities

The 10 genera with the highest relative abundance were selected for clustering heat map analysis (Figure 5). In terms of bacterial community composition, DA1, DA3, and DA7 were separated from each other, indicating that the microbial communities of dry-aged beef with different aging times varied greatly. Moreover, two distinct clusters were clearly observed; cluster Ⅰ consisted of DA1 and WA1 samples, and cluster Ⅱ consisted of CG, WA3, and WA7. On the contrary, the fungal communities of wet-aged beef varied greatly due to the separate distribution of WA1, WA3, and WA7. Notably, two distinct clusters were also observed; WA1 and DA3 had a high degree of similarity, DA1, DA7, and WA3 were clustered together, while CG belonged to a separate group and was farthest from the other groups. The results showed that the aging time and method had a significant effect on the composition and distribution of microbial communities in beef. Setyabrata et al. [10] also found a clear separation in the diversity of microbial community composition between dry- and wet-aging methods.

### 3.5. Correlation Analysis between Dominant Microbiota and Key Volatile Compounds

In this study, the PLSR method combined with a correlation network model was applied to analyze the relationship between eleven key volatile compounds (with OAV >1) and twenty dominant microorganisms (with the top 10 bacterial and fungal genera in relative abundance, respectively). As presented in Figure 6A, the X-matrix was designed as the core microorganisms, the Y-matrix was set as key volatile compounds, and the active matrix was represented by different aging samples. The PLSR results showed that the first two components (t1 and t2) could mainly explain the X (R^2^X cum = 0.940) and Y (R^2^Y cum = 0.612) variables. Obviously, beef wet aging for 7 d (WA7) was significantly and positively correlated with *Prauserella*, *Trichosporon*, 2-decenal, (E)-, 1-octen-3-ol, hexanal, octanal, 1-pentanol, 2-octenal, (E)-, and heptanal. In addition, the samples of CG and DA1 were positively correlated with *Brochothri*, *Rhodotorula*, *Malassezia*, *Talaromyces*, and cis-4-decenal. The PLSR analysis revealed that DA7 samples had a significant correlation with *Rhodococcus*, *Paenisporosarcina*, and 2,4-Decadienal, (E,E)-. Additionally, the observed changes in flavor compounds and microorganisms in different aging samples could provide useful information on the mechanism of flavor formation in aged beef.

The variable importance for projection (VIP) calculation was used to describe the strength and explanatory power of microbial influence on volatile compound formation (Figure 6B). A high correlation was deemed as the absolute value of the linear correlation coefficient |ρ| was over 0.7 [39], and a variable with a VIP value over 1.0 [40]. Microorganisms that met the above conditions were considered the dominant components of the microflora responsible for flavor formation [32]. As shown in Figure 6C, the five genera (*Trichosporon*, *Prauserella*, *Rhodotorula*, *Malassezia*, and *Corynebacterium*) constituting the microflora could be selected as functional microbiota associated with flavor formation. It was easily observed that *Trichosporon* was beneficial in promoting the formation of nine key volatile compounds in aged beef, while *Rhodotorula*, *Malassezia*, and *Corynebacterium* were negatively correlated with some volatile components. Therefore, these dominant microbial genera could be considered the markers of the microbial community to characterize the aged beef and the formation mechanism of flavor components.

## 4. Conclusions

This present study provides information on the relationship between dominant microbiota and key volatile compounds in aged beef via correlation analysis. Post-mortem processes had a significant effect on the apparent meat quality, flavor composition, and microbial communities of beef. Compared to dry-aged beef, wet-aged beef within 7 d has higher product yield, more stable color, and enriched flavor profiles. PLSR and the correlation network model showed that *Trichosporon*, *Prauserella*, *Rhodotorula*, *Malassezia*, and *Corynebacterium* were strongly associated with the formation of flavor substances. Among them, *Trichosporon* plays an important role in the flavor development of aged beef as evidenced by a positive correlation with most of the key volatile compounds, including 1-octen-3-ol, hexanal, octanal, 1-pentanol, 2-octenal, (E)-, heptanal, and 2-decenal, (E)-. These findings can provide a rationale and guideline for the quality control of different post-mortem aging beef via the regulation of functional microbiota.

## Figures and Tables

**Figure 1 foods-12-03266-f001:**
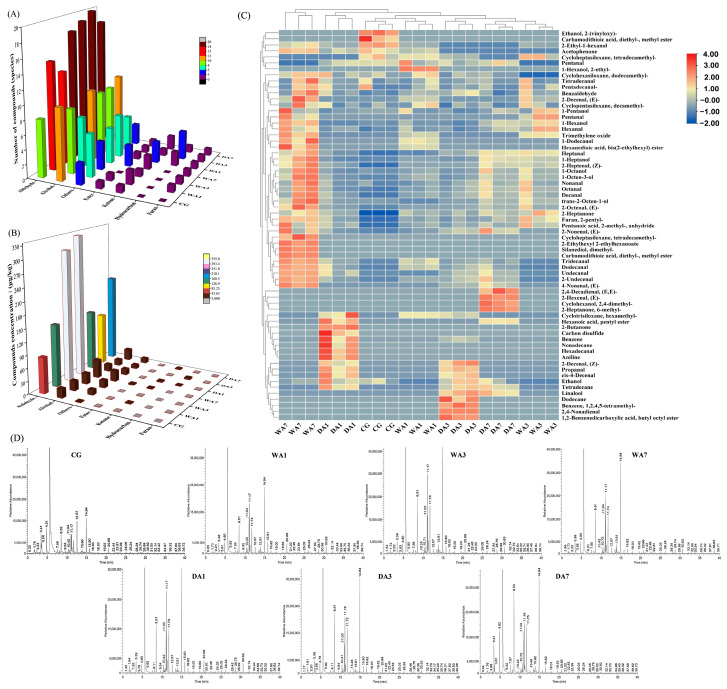
Visualization analysis of the volatile compounds in beef during postmortem aging based on GC–MS data. (**A**,**B**) Number and concentration of various species of volatile compounds. (**C**) Heatmap and cluster analysis of volatile compounds. (**D**) Total ion chromatogram of volatile compounds. Note: WA1~WA3, wet aging 1 d~7 d; DA1~DA7, dry aging 1 d~7 d.

**Figure 2 foods-12-03266-f002:**
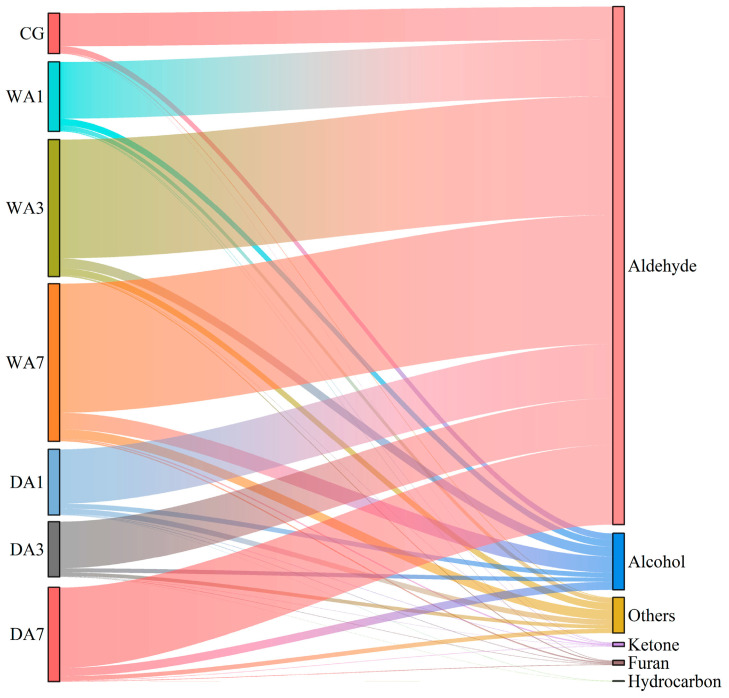
Sangji map of volatile compound species.

**Figure 3 foods-12-03266-f003:**
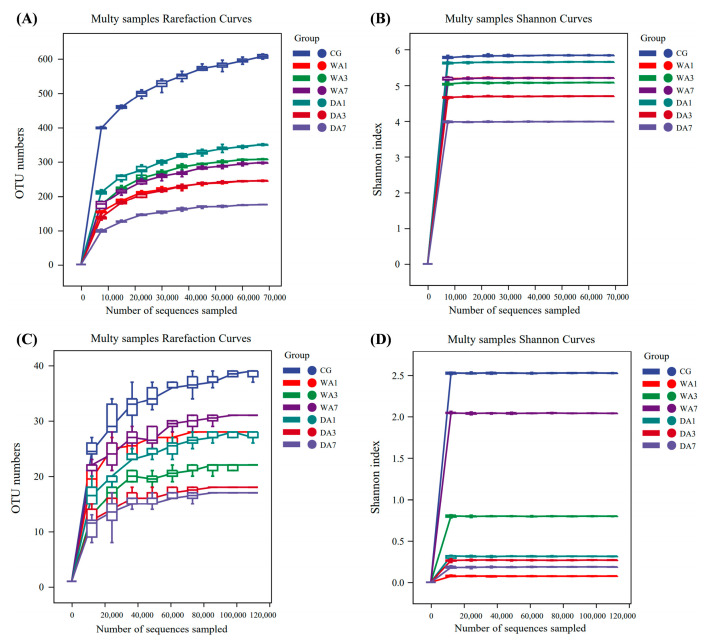
The Rarefaction and Shannon curves of bacteria (**A**,**B**) and fungi (**C**,**D**).

**Figure 4 foods-12-03266-f004:**
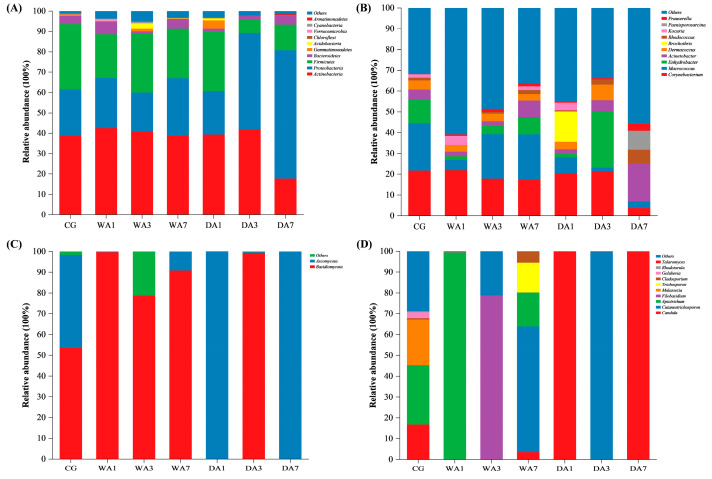
Microbial community composition of beef during wet and dry aging. (**A**,**B**) Relative abundance of bacteria in beef at the phylum and genus level; (**C**,**D**) Relative abundance of fungi in beef at the phylum and genus level.

**Figure 5 foods-12-03266-f005:**
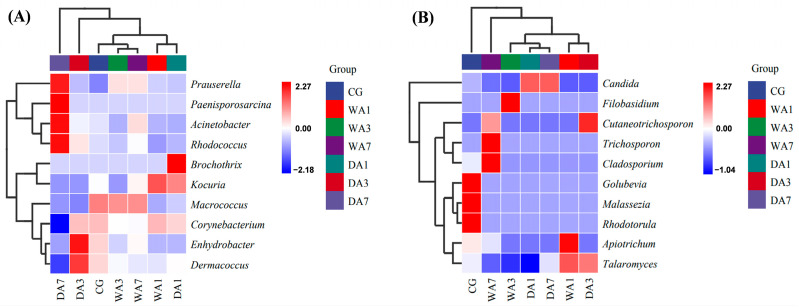
Species composition cluster heatmap of microbial community composition of beef rump during wet and dry aging. (**A**) Species composition of bacteria; (**B**) Species composition of fungi.

**Figure 6 foods-12-03266-f006:**
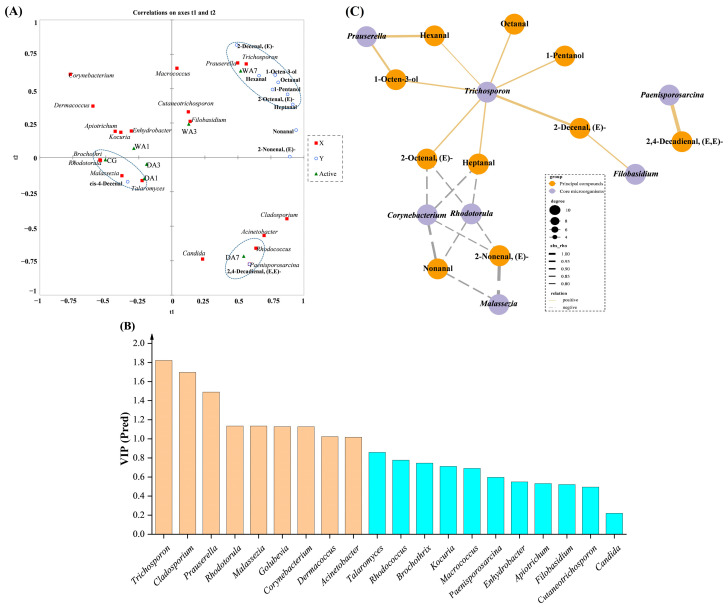
(**A**) Correlation loading of the PLSR analysis between eleven key volatile compounds and twenty dominant microorganisms. (**B**) VIP scores and 95% confidence interval of microbial community at the genus level. (**C**) Visual correlation network of dominant Microbiota and the identified key volatile compounds in aged beef.

**Table 1 foods-12-03266-t001:** Changes in pH, aging loss, and color of beef during wet and dry aging.

Traits	CG	WA1	WA3	WA7	DA1	DA3	DA7
pH	6.05 ± 0.01 ^a^	5.70 ± 0.03 ^cd^	5.71 ± 0.05 ^cd^	5.88 ± 0.09 ^b^	5.62 ± 0.02 ^d^	5.73 ± 0.03 ^c^	5.81 ± 0.03 ^b^
Aging loss (%)	-	0.42 ± 0.15 ^e^	3.43 ± 1.33 ^d^	5.04 ± 1.44 ^c^	6.11 ± 0.45 ^c^	14.88 ± 2.11 ^b^	34.27 ± 2.54 ^a^
L*	32.56 ± 0.03 ^d^	36.90 ± 0.05 ^a^	35.63 ± 0.52 ^b^	33.27 ± 0.38 ^c^	36.02 ± 0.07 ^b^	36.52 ± 0.03 ^a^	30.86 ± 0.19 ^e^
a*	16.35 ± 0.19 ^e^	19.49 ± 0.15 ^b^	18.31 ± 0.05 ^c^	17.13 ± 0.35 ^d^	23.45 ± 0.31 ^a^	23.01 ± 0.04 ^a^	15.32 ± 0.47 ^f^
b*	3.44 ± 0.17 ^d^	5.04 ± 0.01 ^b^	3.94 ± 0.04 ^c^	3.30 ± 0.13 ^d^	11.32 ± 0.37 ^a^	11.05 ± 0.11 ^a^	2.56 ± 0.03 ^e^
ΔE	-	5.59 ± 0.12 ^b^	3.68 ± 0.44 ^c^	1.12 ± 0.33 ^e^	11.16 ± 0.47 ^a^	10.86 ± 0.11 ^a^	2.21 ± 0.28 ^d^

Note: WA1~WA3, wet aging 1 d~7 d; DA1~DA7, dry aging 1 d~7 d; Different lowercase letters indicate significant differences for each parameter in each column (*p* < 0.05).

**Table 2 foods-12-03266-t002:** Key volatile compounds (OAV ≥ 1) of beef during wet and dry aging.

RI(DB-5)	Flavor Compounds	Threshold (μg/kg) ^a^	Odor Activity Value (OAV)							Odor Description ^b^	Identification ^c^
			CG	WA1	WA3	WA7	DA1	DA3	DA7		
759	1-Pentanol	0.15	31.42	40.33	69.20	77.87	21.40	17.60	42.27	fruit	MS, RI
982	1-Octen-3-ol	1	2.49	5.70	7.03	13.99	4.27	3.89	6.92	mushroom	MS, RI
802	Hexanal	4.5	15.27	26.68	54.75	54.51	24.47	19.14	34.66	grass, tallow, fat	MS, RI
1104	Nonanal	1	7.85	9.74	17.88	30.93	10.79	12.32	18.32	fat, citrus, green	MS, RI
903	Heptanal	3	1.06	1.76	4.68	5.43	2.33	3.07	4.64	fat, citrus	MS, RI
1006	Octanal	0.7	3.30	7.51	12.49	18.25	6.93	8.42	12.36	fat, soap, lemon, green	MS, RI
1162	2-Nonenal, (E)-	0.08	2.46	5.96	6.88	21.33	7.33	12.29	18.96	cucumber, fat, green	MS, RI
1181	2-Octenal, (E)-	0.2	ND	2.75	6.78	16.55	2.97	3.10	5.22	roast, fatty	MS, RI
1380	2-Decenal, (E)-	0.3	ND	0.11	1.23	2.67	ND	ND	ND	metal, green	MS, RI
1200	cis-4-Decenal	0.04	ND	ND	ND	ND	3.33	2.75	ND	green, must	MS, RI
1284	2,4-Decadienal, (E, E)-	0.07	ND	ND	ND	ND	ND	ND	1.57	seaweed	MS, RI

Note: ^a^ Odor thresholds in water from Ref. ^b^ Odor description comes from https://www.flavornet.org/ (accessed on 6 March 2023); ^c^ Identification based on the Mass spectrum and Kovats retention index; ND: indicates the compounds were not detected at this sample.

**Table 3 foods-12-03266-t003:** Alpha diversity of microorganisms of beef during wet and dry aging.

Group	Bacteria						Fungi					
	Reads	ObservedSpecies	Shannon	Simpson	Chao1	Goods Coverage	Reads	ObservedSpecies	Shannon	Simpson	Chao1	Goods Coverage
CG	166,569	607	5.83	0.93	687.35	0.9986	166,509	39	2.52	0.79	39.18	1.0000
WA1	82,008	245	5.21	0.96	248.09	0.9998	115,829	28	0.07	0.01	27.90	1.0000
WA3	77,527	307	5.07	0.93	309.50	0.9998	152,735	22	0.79	0.34	21.93	1.0000
WA7	82,708	298	5.20	0.94	304.14	0.9996	136,653	31	2.04	0.66	31.15	1.0000
DA1	91,892	351	5.65	0.95	361.44	0.9995	159,339	27	0.31	0.08	27.40	1.0000
DA3	71,420	245	4.70	0.91	245.18	0.9999	148,282	18	0.26	0.06	17.90	1.0000
DA7	71,308	175	3.98	0.91	175.20	0.9999	139,175	17	0.18	0.04	17.05	1.0000

## Data Availability

The data will be made available upon request.

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
