# Peer review of "Characterization and Correlation of Dominant Microbiota and Flavor Development in Different Post-Mortem Processes of Beef"

_foods, 2023, doi:10.3390/foods12173266_

Round 1
Reviewer 1 Report
In the paper "Characterization and Correlation of Dominant Microbiota and 2 Flavor Development in Different Post-mortem Processes of 3 Beef"
In my opinion, it is a good study, the comments and questions are as:
1- Since this article is based on the quantitative of meat flavorings, the relevant information about their analysis should be given in the details of abstract.
2-What is the internal standard and its concentration?
3-Since the standards of the analytes in the study have not been used to identify and determine, the method must be validated and the accuracy and precision of the method should be determined to ensure the acceptability of the results.
4-It should be determined the protocol and reference, for sampling and laboratory conditions of sample.
5-Since in this study didn't use the standards as well as SIM mode and just used the library of GC-MS for identification, it might miss some low concentrations of analytes which may so important indicators of meat quality.
6- The conclusion section should be rewritten based on the results of the study.
Author Response
Thanks so much for your advice on our manuscript. We have carefully studied the comments carefully and tried our best to revise our manuscript according to the comments. Please see the attachment for detailed responses.

Reviewer 2 Report
This study aims to analyze the potential correlation between microbial communities and volatile flavor changes in beef during aging. Generally, topic is essential. The authors should address following points;
1) page 3 line 120-127
The volatile compounds of samples were analyzed by HS-SPME-GC-MS method ac- 121 cording to the method of AlDalali with minor modifications [16]. A two gram sample was 122 weighed into the 10 mL glass bottle, and 1 µL of internal standard (2-methyl-3-heptanone, 123 0.816 mg/L) and 1 µL of saturated table salt water were added, separately. The vial was 124 sealed with a silicone septum and equilibrated in a water bath at 80°C for 50 min. Simul- 125 taneously, volatiles were thermally desorbed at 250°C for 5 min in the GC-MS injection 126 port in splitless mode. Chemical separation was performed individually on DB-5MS col- 127 umns (30 m × 0.25 mm × 0.25 μm; Agilent Technology, USA) The chromatographic con- 128 ditions were as follows: Nitrogen was used as the carrier gas at a flow rate of 2 mL/min,...
It seems that flow rate of 2mL is too high when 0.25mm i.d capillary column use
2) page 7 line 286-297
....In the present study, the odor activity value (OAV) was used as an important indica- 287 tor to identify the key volatile compounds during beef aging. As shown in Table 2, eleven 288 volatile compounds had OAVs greater than the threshold value. 1-Pentanol had the high- 289 est average OAV, followed by hexanal, nonanal, and 2-nonenal, (E)-. Moreover, octanal 290 and 1-octen-3-ol were found to contribute significantly to the overall aroma of beef, ex- 291 hibiting characteristic aroma notes such as mushroom, fatty, grassy, citrus, soapy, honey, 292 apple peel and fruity [30]. These findings corroborate the results reported by Setyabrata 293 et al. [10]. The key volatile compounds began to form in the early stages of post-mortem 294 aging, and most of the compounds enriched with increasing aging time, especially in WA7 295 samples...
The authors should give tandem MS spectra, including TIC in order to increase article quality.
3) page 9 line 320-333
This part was also given as a table to compare the results with the literature, which shows that the article contribution.
minor revision is needed
Author Response

(The authors gave the same response as above.)
